# The Longitudinal Dividing Bacterium *Candidatus* Thiosymbion Oneisti Has a Natural Temperature-Sensitive FtsZ Protein with Low GTPase Activity

**DOI:** 10.3390/ijms23063016

**Published:** 2022-03-10

**Authors:** Jinglan Wang, Silvia Bulgheresi, Tanneke den Blaauwen

**Affiliations:** 1Bacterial Cell Biology and Physiology, Swammerdam Institute for Life Science, University of Amsterdam, 1098 XH Amsterdam, The Netherlands; j.wang3@uva.nl; 2Environmental Cell Biology, Department of Functional and Evolutionary Ecology, University of Vienna, Djerassiplatz 1, 1030 Vienna, Austria; silvia.bulgheresi@univie.ac.at

**Keywords:** bacterial cytoskeleton, cell division, FtsZ assembly, temperature-sensitive

## Abstract

FtsZ, the bacterial tubulin-homolog, plays a central role in cell division and polymerizes into a ring-like structure at midcell to coordinate other cell division proteins. The rod-shaped gamma-proteobacterium *Candidatus* Thiosymbion oneisti has a medial discontinuous ellipsoidal “Z-ring.” *Ca.* T. oneisti FtsZ shows temperature-sensitive characteristics when it is expressed in *Escherichia coli*, where it localizes at midcell. The overexpression of *Ca.* T. oneisti FtsZ interferes with cell division and results in filamentous cells. In addition, it forms ring- and barrel-like structures independently of *E. coli* FtsZ, which suggests that the difference in shape and size of the *Ca.* T. oneisti FtsZ ring is likely the result of its interaction with Z-ring organizing proteins. Similar to some temperature-sensitive alleles of *E. coli* FtsZ, *Ca.* T. oneisti FtsZ has a weak GTPase and does not polymerize in vitro. The temperature sensitivity of *Ca*. Thiosymbion oneisti FtsZ is likely an adaptation to the preferred temperature of less than 30 °C of its host, the nematode *Laxus oneistus.*

## 1. Introduction

Bacterial cell division is achieved through a large protein complex at midcell, termed the divisome, that is responsible for the synthesis of the envelope of the new cell poles. Assembly of the divisome is initiated by the FtsZ protein, which is a tubulin homologue distributed in the cytoplasm of most bacteria [1]. FtsZ was named after the phenotype of its mutant that forms long filaments in which genome replication and chromosome segregation continues without division. Hence, the related genes were referred to as *fts* (filamentous temperature-sensitive) [2]. FtsZ assembles into protofilaments to form a ring-like structure at midcell, which follows the cell circumference and is stabilized, and bound to the cytoplasmic membrane by early divisome proteins. This "proto-ring" functions as a scaffold for the recruitment of the downstream septal synthesizing components of the divisome and coordination of cytokinesis [3].

FtsZ is composed of two globular subdomains separated by a central core helix, H7 [1]. It has weak sequence homology to tubulin but a similar tertiary structure and tubulin signature motif GGGTGS/TG (in the T4-loop) [4]. In vitro, FtsZ polymerizes into protofilaments by the head-to-tail association of individual subunits. FtsZ can assemble into a range of polymeric forms from filaments to rings, bundles, tubules, sheets, and curves, depending on the different conditions. The Z-ring membrane anchor ZipA protein promotes a large, bundled network of FtsZ polymers in vitro [5]. Z-ring-associated proteins ZapA, ZapC, and ZapD also promote bundles of FtsZ in vitro [6]. As a self-activating GTPase, the polymerization of FtsZ depends on GTP binding but not hydrolysis [7]. GTP, GDP, and other GTP analogs, such as GTP-γ-S and GMPPNP, can bind FtsZ and induce FtsZ polymerization [8]. The behavior of most FtsZ assemblies is characterized by a cooperative assembly in the presence of GTP. In cooperative polymer assembly, the degree follows a sigmoidal time course and exhibits a sharp transition above the critical concentration [9,10]. The critical concentration for single-stranded linear FtsZ polymers and other higher-order structures differ depending on the buffer composition (various monovalent cations, crowding agents, and polycations), and on temperature and pH [3,11]. Single-stranded *E. coli* FtsZ protofilament formation was observed above the critical concentration of 0.88 ± 0.25 μM by cryo-electron microscopy. Interestingly, the concentration of FtsZ in vivo exceeds the critical concentration fivefold [12], implying that it will be able to polymerize all the time. Consequently, premature ring-formation is inhibited by various counteracting systems in *E. coli* [13].

However, the arrangement of the FtsZ filaments in vivo is still not well understood. The association between FtsZ and the inner membrane (IM) is essential for FtsZ assembly. The widely conserved actin-like membrane binding protein FtsA serves as a principal anchor to tether FtsZ to the IM [14,15]. In addition to FtsA, ZipA is another membrane tether for FtsZ and is present only in Gamma-proteobacteria, and its function was shown to be partially redundant when FtsA is present [16,17]. Many nonessential proteins, such as ZapA, C, D, E, and G, interact with FtsZ to regulate the structure or stability of the Z-ring and to use it as a scaffold to enable interaction with other proteins such as ZapB [1,18]. The super-resolution fluorescence microscopy images of Z-rings in *Escherichia coli*, *Bacillus subtilis*, *Staphylococcus aureus*, *Caulobacter crescentus,* and *Streptococcus pneumoniae* revealed that FtsZ filaments form a discontinuous and clustered Z-ring of relatively short filaments [19]. The Z-ring is highly dynamic in vivo and exhibits a rapid exchange of subunits with the FtsZ molecules in the cytoplasmic pool, which depends on its GTP hydrolysis activity [20,21]. The treadmilling dynamics of FtsZ polymers were first discovered in 2017 through a functional Fluorescent Protein (FP)-FtsZ-fusion and revealed directional movement by the continuous addition of new subunits at the front of the polymer concurrent with the release of old subunits at the opposite end [22]. This motion drives peptidoglycan synthesis at the septum, building a progressively smaller ring to constrict and divide cells. FtsZ filaments assembled on lipid bilayers confirmed treadmilling dynamics in vitro, demonstrating that this phenomenon is intrinsic to FtsZ polymers [23].

In this study, we characterized the FtsZ protein of the longitudinal dividing bacterium *Candidatus* Thiosymbion oneisti. *Ca.* T. oneisti is a yet uncultivable rod-shaped gamma proteobacterium. As the ectosymbiont of the nematode host *Laxus oneistus*, it lives on its cuticle and forms a palisade-like single-layer perpendicular to the surface. Phylogenomics, transcriptomics, and proteomics, as well as stable isotope-based techniques, indicated that free-living sulfur-oxidizing purple bacteria are the closest relative to the genus *Candidatus* Thiosymbion and that *Ca.* T. oneisti is a chemo-heterotroph [24,25]. Rod-shaped cells usually grow in length and place their Z-ring at midcell, perpendicular to their long axis. However, *Ca.* T. oneisti grows in width instead of elongating and divides along its long axis (medially). Its Z-ring is elliptical, discontinuous, and medial [26]. In addition, its actin homolog (MreB) forms a medial ring-like structure at the cell periphery throughout the cell cycle and is required for Z-ring formation [27]. In most investigated rod-shaped bacteria, MreB is used to recruit a protein complex termed the elongasome and to direct synthesis of the cylindrical part of the envelope [28]. The peptidoglycan layer of *Ca.* T. oneisti is more cross-linked than that of the well investigated Gram-negative bacterium *E. coli*, but has relatively short glycan chains with an average length of 13 disaccharides [29]. To further investigate FtsZ of *Ca.* T. oneisti (hereafter referred to as FtsZ^TO^) and understand how the elliptical ring of FtsZ^TO^ can achieve its function in vivo, we examined the morphological effects of overexpression and localization of FtsZ^TO^ in *E. coli*. Furthermore, we purified the FtsZ^TO^ protein and measured its GTPase hydrolysis and polymerization properties in vitro. Our results demonstrate that FtsZ^TO^ can form ring structures independently in *E. coli* cells and that its behavior is sensitive to temperature. In vitro, FtsZ^TO^ showed a low GTPase activity and did not exhibit a significant polymerization ability. 

## 2. Results

### 2.1. FtsZ^TO^ Interferes with Cell Division in E. coli and Overexpression of FtsZ^TO^ Results in Filamentous Cells

To characterize FtsZ^TO^ in more detail, we first compared the morphological effects of the overexpression of FtsZ^TO^ and FtsZ of *E. coli* (hereafter referred to as FtsZ^EC^) on *E. coli*, which were separately expressed from plasmids under control of the IPTG-inducible *pTrc-down* promoter. FtsZ^TO^ expression in wild-type *E. coli* grown in TY medium induced by 15 μM IPTG at 28 °C for 4 h resulted in a cell length increase, whereas the FtsZ^EC^ expression did not show any significant morphological changes under the same conditions (Figure 1a). A higher-level induction (50 μM IPTG) caused a similar degree of filamentation in cells expressing FtsZ^TO^ or FtsZ^EC^ (Figure 1b). As expected, an increase in cell length of *E. coli* was observed with an increase in overproduction of FtsZ^TO^ (Figure 1b). An amount of 15 μM IPTG induced the production of about the same amount of protein as the endogenous FtsZ, as shown by Western blot analysis (Figure 1c). As the same amount of FtsZ^EC^ did not affect the cell length, the cell length increase upon expression of FtsZ^TO^ indicates that the hybrid FtsZ^TO^/FtsZ^EC^ situation did not fully support cell division. When FtsZ^TO^ and FtsZ^EC^ were overproduced by adding 50 μM IPTG, both cells became filamentous because the concentration of FtsZ was high enough to cause filamentation due to the titration of other divisome proteins [30,31].

### 2.2. FtsZ^TO^ Colocalizes with FtsZ^EC^ Rings in E. coli

To determine whether the filamentation was caused by the presence of FtsZ^TO^ in the ring or because it was titrating FtsZ^EC^ away from the ring, we constructed a N-terminal fluorescent fusion of FtsZ^TO^ (mCherry-FtsZ^TO^). Although mCherry-FtsZ^TO^ did not show the same cell length increase by 15 μM IPTG induction in *E. coli* as the original FtsZ^TO^, the overproduction of mCherry-FtsZ^TO/EC^ under higher IPTG (50 μM) induction produced an identical filamentous phenotype (Figure 2a). Midcell rings of mCherry-FtsZ^TO^ were visible in dividing cells, indicating that it can be part of the Z-ring (Figure 2b). As the mCherry-FtsZ^EC^ signal showed a higher fluorescent background and formed more multiple rings along cells compared to mCherry-FtsZ^TO^ (Figure 2b), we hypothesize that mCherry-FtsZ^TO^ interacts with native FtsZ^EC^ in *E. coli* to form the ring but is not under *E. coli* Z-ring’s dynamic spatiotemporal regulation. To examine this hypothesis, mCherry-FtsZ^TO^ and mTurquoise2 (mTq2) -FtsZ^EC^ were co-transformed, and their localization was analyzed. When induced with 15 μM IPTG, the cells formed filaments and the complete colocalization of mCherry-FtsZ^TO^ and mTq2-FtsZ^EC^ was observed (Figure 2c), which supports the hypothesis that ectopically expressed FtsZ^TO^ and FtsZ^EC^ may polymerize together to form a ring. 

### 2.3. FtsZ^TO^ Forms Ring- and Barrel-like Structures Independently of FtsZ^EC^ in the E. coli FtsZ Depletion Strain VIP205

To investigate whether FtsZ^TO^ would be able to polymerize into a ring without FtsZ^EC^, plasmids pJW15 and pJW16 were constructed that expressed both FtsZ^TO^ and mCherry-FtsZ^TO^ in the same operon or only mCherry-FtsZ^TO^ under the control of the arabinose-inducible promoter, respectively. As the fluorescent fusions to FtsZ are, in general, not fully functional, mCherry-FtsZ^TO^ is only able to polymerize into a ring in the presence of untagged FtsZ^TO^ (Figure 3). The plasmids were transformed into strain VIP205, in which the native FtsZ^EC^ can be depleted as the *ftsZ* gene is dissociated from its native promoter and expressed from the chromosome by an IPTG-inducible promoter [32]. VIP205 carrying pJW15 or pJW16 exhibited a normal rod-shape morphology and midcell localization of mCherry-FtsZ^TO^ (Figure 3) when grown in TY medium in the presence of 0.02% arabinose and 10 μM IPTG at 28 °C for 4 h; the cells were similar to wild-type cells expressing mCherry-FtsZ^TO^ from pJW13, as shown in Figure 2. When pJW15 was induced with arabinose in the absence of IPTG for 4 h, cells became extremely filamentous and FtsZ^TO^ assembled into multiple rings, indicating that it is likely able to polymerize on its own (Figure 3). A longer time of induction led to regularly and evenly distributed barrel-shaped fluorescence signals (Figure 3). We used the FtsZ^EC^ antibody on Western blots to detect the amount of FtsZ^EC^ and FtsZ^TO^ (Appendix A). However, after IPTG was removed for 6 h, there was still a very small amount of FtsZ^EC^ left in the cells, and FtsZ^TO^ and mCherry-FtsZ^TO^ were almost equally expressed after induction. As an essential gene, *ftsZ^EC^* cannot be deleted in *E. coli*. When only mCherry-FtsZ^TO^ from pJW016 was induced with 0.02% arabinose in VIP205 for 6 h, no FtsZ^TO^ rings or barrels were observed any more (Figure 3). Obviously, not enough FtsZ^EC^ was present to assist in the formations of the FtsZ^TO^ rings observed when mCherry-FtsZ^TO^ and FtsZ^TO^ were both present. In conclusion, the comparison of FtsZ^TO^ and mCherry-FtsZ^TO^, and mCherry-FtsZ^TO^ only, revealed that in *E. coli*, FtsZ^TO^ can independently self-assemble into ring structures on the cytoplasmic membrane, although the FtsZ^TO^ ring is not fully functional and not under proper cell cycle control.

### 2.4. GTPase Activity Assay of FtsZ^TO^ In Vitro

As FtsZ^TO^ appeared to be active in *E. coli,* we performed a GTPase activity assay to investigate the FtsZ^TO^ behavior in vitro. SDS–PAGE and Western blot of purified FtsZ^TO^ confirmed the absence of FtsZ^EC^ contamination (Appendix A). A quantitative amino acid analysis of FtsZ^TO^ was performed to calibrate against a commercial Bradford assay for concentration determination. Circular dichroism spectra for FtsZ^EC^ and FtsZ^TO^ were identical, suggesting a similar structure for both proteins (Appendix A). The GTPase of FtsZ^TO^ was measured in MES buffer pH 6.5 at 28 °C (Figure 4a and Table 1). However, FtsZ^TO^ showed a low GTPase activity compared to FtsZ^EC^ (Appendix A). Unlike FtsZ^EC^ and many other FtsZ proteins, the GTPase activity of FtsZ^TO^ under lower pH conditions gave a small increase of 5% instead of a decrease (Figure 4b). Two mutants of the bottom interface of FtsZ were selected as negative controls to confirm the GTPase activity of FtsZ^TO^. Mutations I207M of the T7 loop and I273E of the H10 helix mutations reduced the GTPase activity by ~65% and ~90%, respectively, in HEPES buffer pH 7.5 and exhibited a higher GTPase activity in MES buffer pH 6.5 than in HEPES buffer pH 7.5, as observed with wild-type FtsZ^TO^ (Figure 4b). 

### 2.5. FtsZ^TO^ Fails to Polymerize but Interacts with MreB^TO^ In Vitro

We investigated the polymerization potential of purified FtsZ^TO^ in vitro by 90°-angle light scattering and electron microscopy, which are two standard assays for FtsZ assembly. However, the light scattering signal of FtsZ^TO^ remained at the baseline at 28 °C and no clear FtsZ^TO^ protofilaments were visible by EM in HEPES buffer or MES buffer and 5 mM MgCl_2_ with up to 10 mM GTP (Figure 5), indicating that FtsZ^TO^ failed to polymerize in vitro. The sedimentation assay by ultracentrifugation showed the same result as light scattering (Appendix A). Extra Mg^2+^ and Ca^2+^ promoted FtsZ^TO^ aggregation and enhanced the light scattering signal baseline. This defect of FtsZ^TO^ assembly dynamics is consistent with what has been reported for FtsZ mutants with a low GTPase activity [21]. 

MreB^EC^ and FtsZ^EC^ are reported to colocalize at midcell in vivo and directly interact in vitro, as shown in a bacterial two-hybrid screen [33,34]. The MreB^EC^ D285A variant abolishes the interaction with FtsZ^EC^ but not with components of elongasome [34]. MreB^TO^ also colocalizes with FtsZ^TO^ and accumulates around midcell before the FtsZ^TO^-ring mediates divisome assembly [27]. To test the interaction of MreB^TO^ and FtsZ^TO^ in vitro, we performed a Ni-NTA pull-down assay using His-tagged MreB^TO^ as a bite protein. A SUMO tag was fused to the amino terminus of MreB^TO^ to promote expression level and stability [35], which also increased the molecular mass of MreB^TO^ to distinguish easily between FtsZ^TO^ and MreB^TO^ in SDS–PAGE. FtsZ^TO^ was shown to have a weak interaction with His-SUMO-MreB^TO^, and this interaction was strongly reduced in the corresponding variant MreB^TO^ D285A (Figure 6).

### 2.6. FtsZ^TO^ Shows Temperature-Sensitive Characteristics in E. coli

*L. oneistus* is not resistant to heat stress and its survival rate drops significantly from 90% to 50% (after three hours of incubation at 41 °C or 42 °C, respectively) [36]. Hence, we investigated the effects of temperature on FtsZ^TO^. The overexpression of FtsZ^TO^ with 50 μM IPTG induction for six mass doubling times caused filamentation of wild-type *E. coli* cells at 28 °C (around ambient temperature of *L. oneistus* and *Ca.* T. oneisti) and at 37 °C. Interestingly, cells at 42 °C became rounded and could not maintain their rod shape after induction (Figure 7). The midcell localization of mCherry-FtsZ^TO^ in wild-type *E. coli* was only present when induced at 28 °C. At 37 °C and 42 °C, mCherry-FtsZ^TO^ lost its localization and some cells formed inclusion bodies at 42 °C. For VIP205 cells carrying pJW15-expressing FtsZ^TO^ and mCherry-FtsZ^TO^ induced for 6 h with 0.02% arabinose, clear rings and barrel structures were formed at 28 °C and regular patches were observed at 37 °C. No clear structures were observed in cells at 42 °C, but some inclusion bodies were produced. These results suggest that the morphological effects of FtsZ^TO^ on *E. coli* and its localization are temperature-sensitive and that FtsZ^TO^ is adapted to function at 28 °C.

## 3. Discussion

### 3.1. FtsZ^TO^ Is Adapted to 28 °C and Temperature Increases Affect Its Localization

The FtsZ protein is highly conserved throughout the Bacteria, the Euryarchaeota, and mitochondria of the Eukarya. Compared to other bacterial species, the longitudinally dividing *Ca.* T. oneisti synthesizes a longer septum and likely spends more energy for septating and constricting from the poles to midcell. Another closely related nematode symbiont, *Ca.* Thiosymbion hypermnestrae, also septates and divides longitudinally, but it does so asynchronously, i.e., from the host-attached pole to the distal-free pole [27]. It is not known whether longitudinal division is caused by the unusual characteristics of the FtsZ protein or by specific regulation of the FtsZ structure by other proteins. To differentiate between these possibilities, we wanted to determine whether FtsZ^TO^ was behaving similarly or differently from the well characterized FtsZ^EC^, which mediates transverse division. The alignment of FtsZ^TO^ and FtsZ^EC^ showed a 61.2% sequence identity and 75.6% sequence similarity (Appendix A). The residues for GTP binding [37,38] are identical in both proteins (Appendix A). Although no crystal structure of FtsZ^TO^ is available, a predicted structure can be built based on existing FtsZ structures in the PDB-data Bank using the online program Phyton2 [39]. The globular core and C-terminal domain (CTD) of the FtsZ^TO^ model most closely resembled the structure of *Pseudomonas aeruginosa* FtsZ (Appendix A) [40]. To obtain a structure that is independent from existing FtsZ structures, the Alphafold v2.0 system [41] was used to predict the structure of FtsZ^TO^. Although these two structures showed differences, the positions of the T7 loop and H10 helix important for the subunit interaction and GTPase activity of two FtsZ subunits were very similar (Appendix A). The T7 loop of FtsZ^TO^ is fairly conserved between FtsZ^EC^ and *Methanococcus jannaschii* FtsZ (FtsZ^MJ^) [42], whereas helix 10 of the various species has differences in amino acid composition and length (Appendix A). This causes potentially small differences in the orientation of the two FtsZ molecules in the dimer. This provides sufficient possibilities for FtsZ^TO^ and FtsZ^EC^ to interact, but likely not to support a GTPase activity at the rate of FtsZ^EC^. FtsZ^TO^ has a longer C-terminal linker (CTL), which is a known variable involved in regulation among FtsZ homologs (Appendix A) [43].

We employed *E. coli* cells to study FtsZ^TO^ as *Ca.* T. oneisti is still uncultivable. Surprisingly, the morphological effects and localization of FtsZ^TO^ in *E. coli* varied from low to high temperature and exhibited sensitivity to heat. The FtsZ^TO^ rings were only observed at 28 °C, which is close to the environmental temperature of the nematode habitat, but not at higher temperatures. Seven temperature-sensitive alleles of FtsZ^EC^ have been described until now: FtsZ84 (G105S), FtsZ26, FtsZ6460 (G109S), FtsZ972 (A129T), FtsZ2066 (V157M), FtsZ9124 (P203L), and FtsZ2863 (A239V) [44]. These mutants can form functional Z-rings at midcell at the permissive temperature of 30 °C, but fail to localize at the nonpermissive temperature of 42 °C. Assembly characteristics of FtsZ in vitro do not always correspond with features in vivo as has been reported [45,46]. For example, some temperature-sensitive FtsZ mutants have measurable low GTPase activities in vitro. None of them exhibit polymerization activity when assayed by light scattering or electron microscopy in vitro, despite being functional in vivo at the permissive temperature [44,45]. Consistent with these temperature-sensitive alleles of FtsZ^EC^, FtsZ^TO^ also displayed a low rate of GTP hydrolysis and no polymerization under different buffers and conditions in vitro (Appendix A). The GTP affinity of FtsZ^TO^ is unlikely to be the reason for its low GTPase activity, given the conservation of its GTP binding and catalytic residues. The GTPase activity of FtsZ is highly coupled with its polymerization activity. Residues in the interface between adjacent monomers, involved in intermolecular interaction, also affect the GTPase activity of FtsZ [47]. FtsZ^TO^ does not contain any of the mutations reported for temperature-sensitive FtsZ^EC^. It remains unclear which residue(s) contribute(s) to the temperature-sensitive biochemical properties of FtsZ^TO^. The preferable temperature for the nematode is below 30 °C and *L. oneistus* cannot survive above 41 °C [36]. Given the FtsZ^TO^ properties, we demonstrate that it makes sense that *Ca.* T. oneisti is a heat-sensitive organism and not resistant to temperatures above 37 °C.

### 3.2. FtsZ^TO^ Can Form Rings in E. coli Independent from FtsZ^EC^

The overexpression of FtsZ in *E. coli* has been reported to interfere with cell division [30]. As expected, the overexpression of FtsZ^TO^ also caused filamentation of *E. coli* cells grown at 28 °C. Compared to native FtsZ^EC^, FtsZ^TO^ affected the cell length at very low expression levels. This phenomenon is probably caused by the FtsZ^TO^ monomer competing with FtsZ^EC^ ones during polymerization. However, FtsZ^TO^ incompetence to hydrolyze GTP affects the full function and dynamics of the FtsZ^EC^ polymer in vivo. Similarly, overexpression of the temperature-sensitive FtsZ^EC^ mutant FtsZ6460 (G109S), an allele with detectable low GTPase activity, led to a dramatic increase in the cell length, blocking normal cell division [44]. When FtsZ^EC^ was twofold-overexpressed (compared to WT level), the extra FtsZ molecules caused a multi-stranded configuration of FtsZ polymers and an increase in the Z-ring toroidal zone affecting Z-ring constriction [48]. Hence, both FtsZ^TO^ and FtsZ^EC^ effectively produced filamentous cells with 50 μM IPTG induction. In contrast to the multiple rings observed in *E. coli* cells overexpressing FtsZ^EC^, the barrel-like structures observed in cells that uniquely express FtsZ^TO^ (VIP205) are probably due to the absence of proper regulation of FtsZ^TO^.

The structure and dynamics of FtsZ polymers are strongly coupled to GTPase activity. Filaments of the FtsZ^EC^ mutant with low GTPase activity have a lower turnover rate and reduced treadmilling speed in vivo [21]. The speed and direction of divisome movement is dependent on and the same as FtsZ treadmilling [22,49]. FtsZ^EC^ GTPase mutants with severely reduced GTP hydrolysis rates (corresponding to severely reduced treadmilling speed) will alter the septum morphology. Notably, it has been proven that peptidoglycan synthesis activity and incorporation rate are independent of FtsZ treadmilling, and FtsZ^EC^ GTPase mutants do not change the rate of peptidoglycan synthesis in vivo [21]. This is probably the reason why some FtsZ^EC^ GTPase mutants with reduced GTPase activity, including temperature-sensitive variants, exhibit robust Z-ring formation and division [21,44,45]. The GTPase activity (Pi/FtsZ/min) of FtsZ^TO^ is about three times higher than that of the temperature-sensitive FtsZ^EC^ allele FtsZ84 (G105S) measured under similar conditions (MES buffer pH 6.5, 1 mM GTP concentration but at a higher temperature of 30 °C) in vitro [45]. We speculate that FtsZ^TO^ has a GTPase activity several times higher than that of FtsZ84 (G105S) in vivo, which should be enough to form a functional dynamic Z-ring. Accordingly, the lower GTPase activity of FtsZ^TO^ compared with FtsZ^EC^ should not influence the formation of functional FtsZ^TO^ rings. Therefore, the fact that FtsZ^TO^ cannot form a functional ring in *E. coli* is likely due to an incompatibility with the *E. coli* regulatory proteins involved in proto-ring formation. 

### 3.3. Ca. T. oneisti Lacks Some of the Z-Associate Proteins

In *E. coli,* the Z-ring is bound by the membrane anchor proteins FtsA/ZipA tethering FtsZ^EC^ to the inner membrane and at least five ring-associated proteins ZapA-E. ZapB and ZapC are absent from the genome of *Ca.* T. oneisti. ZapC prevents depolymerization through inhibition of GTP hydrolysis by FtsZ and promotes the lateral interaction of filaments [50]. The FtsZ^TO^ polymers may be more stable in vivo than FtsZ^EC^ due to weaker GTPase activity. Therefore, the ZapC protein might not be necessary for the divisome of *Ca.* T. oneisti. ZapB works together with ZapA as a complex and persists at midcell, providing additional clues for Z-ring positioning in *E. coli*. MatP, binds in the vicinity of the replication terminus of the chromosome present at midcell and through interaction with ZapA-ZapB, and is involved in the control of Z-ring localization [51,52]. The absence of MatP and ZapB in *Ca.* T. oneisti implies that it does not use this system to regulate chromosomal segregation and Z-ring position. How FtsZ^TO^ determines longitudinal midcell positioning is still unclear. The native FtsZ^TO^ ring in *Ca.* T. oneisti is a large ellipse at initiation of cell division and constricts into a ring during the process of constriction. However, the size and shape of the Z-ring seem to be independent from FtsZ dynamics [53]. Therefore, it is not known how the intrinsic property of FtsZ^TO^ contributes to the organization of the Z-ring.

### 3.4. FtsZ^TO^ Interacts with MreB^TO^

MreB^TO^ is thought to play an essential role in the growth and division of *Ca.* T. oneisti. MreB^TO^ accumulates medially throughout the cell cycle and, therefore, is placed at the prospective division plane even prior to FtsZ^TO^ ring formation. Co-localization of MreB^TO^ and FtsZ^TO^ was observed at the septal plane during cell division, and FtsZ^TO^ rings were barely recognizable when MreB polymerization was blocked. Here, we examined the interaction of MreB^TO^ and FtsZ^TO^ in vitro, and showed that MreB^TO^ binds FtsZ^TO^ in vitro.

### 3.5. Conclusions

To sum up, we propose that FtsZ^TO^ is naturally sensitive to temperature. Purified FtsZ^TO^ has a low GTPase activity, interacts with MreB^TO^, but appears to not be capable of polymerizing in vitro. In *E. coli* cells grown at 28 °C, FtsZ^TO^ can interact with FtsZ^EC^, localizes at midcell, and is likely functional. The overexpression of FtsZ^TO^ resulted in filamentous cells where FtsZ^TO^ assembled into rings and barrels, independently from FtsZ^EC^. 

In conclusion, the behavior of FtsZ^TO^ in *E. coli* cells and the similarity between the two proteins support the hypothesis that the difference in shape and size of the FtsZ^TO^-ring is likely the result of its interaction with Z-ring organizing proteins.

## 4. Materials and Methods

### 4.1. Bacterial Strains and Plasmids

*Laxus oneistus* symbiotic nematodes were collected from a sand bar of Carrie Bow Cay, Belize (16° 48′11.01″ N, 88° 4′54.42″ W), were fixed by methanol and transported deep-frozen [27]. The *E. coli* strains and plasmids used in this study are listed in Table 2. The construction of plasmids is described in Appendix A.

### 4.2. FtsZ^TO^ Expression and Purification

FtsZ^TO^ was isolated from strain BL21DE3 plysS. Cells expressing FtsZ^TO^ from plasmid pJW17 were grown in TY medium (10 g of Tryptone (Bacto Laboratories, Mount Pritchard NSW, Australia), 5 g of yeast extract (Duchefa, Amsterdam, The Netherlands), and 5 g of NaCl (Merck, Kenilworth, NJ, USA) per liter) with 100 µg/mL of ampicillin (Sigma-Aldrich, St. Louis, MO, USA) at 28 °C, and protein expression was induced by 0.5 mM IPTG (Promega, Fitchburg, WI, USA) from OD_600_~0.6 for 6 h. Harvested cells were re-suspended in 50 mM Tris buffer (50 mM Tris-HCl and 1 mM EDTA (Sigma-Aldrich, St. Louis, MO, USA), pH 7.9) with 50 mM KCl and broken by a French Press under a pressure of 800 psi. Cell extracts were separated by ultracentrifugation at 200,000× *g*. Supernatant was loaded onto a 5 mL HiTrap Q HP prepacked column (GE Healthcare, North Richland Hills, TX, USA) using an AKTA system (GE Healthcare, North Richland Hills, TX, USA) that was equilibrated with 50 mM Tris (VWR international, Radnor, PA, USA) buffer with 50 mM KCl (Merck, Kenilworth, NJ, USA) before loading of the samples. The column was washed with 50 mM Tris buffer with 50 mM KCl and 150 mM KCl, successively until the UV signal was stable. FtsZ^TO^ was eluted using a gradient of 150–550 mM KCl in 50 mM Tris buffer, and 1 mL fractions were collected. Purity of the protein was confirmed by 12% SDS–PAGE. FtsZ^TO^ I207M was expressed from plasmid JW20. FtsZ^TO^ I273E was expressed from plasmid JW21. I207M and I273E were isolated from BL21DE3 plysS in the same way as wild-type FtsZ^TO^.

### 4.3. Circular Dichroism

Proteins were in 10 mM NaPO_4_ and 50 mM KCl, pH 7.4, and concentrated/diluted to 0.4 μM. CD measurements were taken using a Jasco J-1500 spectropolarimeter (Jasco, Tokyo, Japan) using a wavelength range of 190–350 nm. The average of 10 runs was taken for each protein with a buffer control subtraction. For a direct comparison, correcting for the differing amino acid sequences, the collected data were converted to molecular CD and plotted against wavelength (nm). The resulting CD spectra are compared in Appendix A and show that FtsZ^TO^ and FtsZ^EC^ have similar CD spectra consisting of both α-helices (~190, 208, and 222 nm) and β-sheets (~210 nm).

### 4.4. Ninety-Degree Perpendicular Light Scattering Assay

A light scattering assay was performed in a quartz cuvette (Hellma Analytics, Müllheim, Germany, 10 mm light path, 4 mm light width) and measured with a spectrofluorimeter (Photon Technology International, Birmingham, NJ, USA) at 28 °C, which is the same temperature used for cell growth. The excitation and emission wavelengths were set to 350 nm. FtsZ^TO^ (25 μM) in HEPES buffer (50 μM HEPES (Sigma-Aldrich, St. Louis, MO, USA), 50 mM KCl, 5 mM MgCl_2_) or MES buffer (50 μM MES (Sigma-Aldrich, St. Louis, MO, USA), 50 mM KCl, 5 mM MgCl_2_) was maintained at 28 °C for 100 s to obtain a baseline, and GTP (Sigma-Aldrich, St. Louis, MO, USA) at the indicated concentration was added into the sample to induce polymerization [8]. 

### 4.5. GTPase Assay

The GTPase activities of FtsZ^TO^ and mutant FtsZ^TO^ were carried out in MES buffer (50 mM MES, 50 mM KCl, 5 mM MgCl_2_, pH 6.5) or HEPES buffer (50 mM HEPES, 50 mM KCl, 5 mM MgCl_2_, pH 7.5) at 28 °C and measured by the Malachite green Phosphate assay kit POMG-25H (BioAssay system, Hayward, CA, USA) as described before [56]. GTP (concentration as indicated) was added into FtsZ (5 μM) for activity determination. 

GTP hydrolysis of FtsZ^TO^ started when GTP was mixed with FtsZ^TO^ in the MES/HEPES buffer and was stopped by the addition of Malachite green. To calculate FtsZ^TO^ GTPase, GTP hydrolysis activity was stopped after 0 min (control sample), 4 min, 5 min, 7 min, 10 min, 15 min, 20 min, and 30 min reaction times. Free phosphate of every sample was calculated using the phosphate standard curve of the assay kit. Released phosphate equals the difference between the free phosphate value of the sample compared to the control sample. GTP hydrolysis rates were calculated from the slopes of the linear increase in released phosphate over time [21,56], and k_cat_ and K_m_ values were obtained from fitting data to the Michaelis–Menten equation (also see Appendix A for an example and Appendix A for all data).

### 4.6. Pull-Down

His-SUMO-MreB^TO^ and His-SUMO-MreB^TO^ D285A were expressed in the BL21DE3 plysS strain from plasmid pJW18/pJW19. Cells were grown in TY medium with 100 µg/mL of ampicillin at 28 °C. Protein expression was induced by 0.5 mM IPTG for 6 h. Harvested cells were re-suspended in 50 mM phosphate-buffered saline at pH 7.4 (PBS) with 500 mM NaCl and broken by a French Press under a pressure of 800 psi. His-SUMO-MreB^TO^ or His-SUMO-MreB^TO^ D285A inclusion bodies were pelleted by centrifugation at 7000× *g* rpm for 20 min. The pellet was washed with 50 mM PBS with 1 M urea (Duchefa, Amsterdam, The Netherlands) and 1% Trition X-100 (Merck, Kenilworth, NJ, USA) for three times. Washed inclusion bodies were resuspended and solubilized in 50 mM PBS with 8 M urea. The solution was centrifuged at 7000× *g* rpm for 20 min and the supernatant was loaded on a 1 mL gravity flow column with Ni-NTA agaroses (QIAGEN, Hilden, Germany) for 10 min. The Ni-NTA column was washed by 20 mL of 50 mM PBS with 50 mM imidazole and 20 mL of 50 mM PBS with urea gradient from 8 M to 0 M urea, and the flow-through solution was discarded. Subsequently, purified FtsZ^TO^ (in 50 mM phosphate-buffered saline, 500 mM NaCl, pH 7.4) was incubated with the Ni-NTA column for 10 min. The Ni-NTA column was washed by another 20 mL of 50 mM PBS with 50 mM imidazole and 500 mM NaCl. Proteins were eluted by loading 1 mL of 50 mM PBS with 300 mM imidazole (Merck, Kenilworth, NJ, USA) and 500 mM NaCl, and interactions were identified by Coomassie blue coloration in 12% SDS-PAGE.

### 4.7. Microscopy and Image Analysis

Cells were fixed by 2.8% formaldehyde and 0.04% glutaraldehyde. Cell suspensions were placed on a 1% agarose layer on glass microscope slides. Fluorescence microscopy images were photographed by a CoolSnap *fx* (photometrics) CCD camera mounted on an Olympus BX-60 microscope with a 100X/N.A. 1.35 oil objective (Olympus, Tokyo, Japan). Images were taken by the program ImageJ [57] with MicroManager (https://micro-manager.org, accessed on 8 March 2022). The localization pattern was analyzed using the public domain program ImageJ [57] in combination with plugin ObjectJ (https://sils.fnwi.uva.nl/bcb/objectj/, accessed on 8 March 2022) and a modified version of Coli-inspector [27,58].

## Figures and Tables

**Figure 1 ijms-23-03016-f001:**
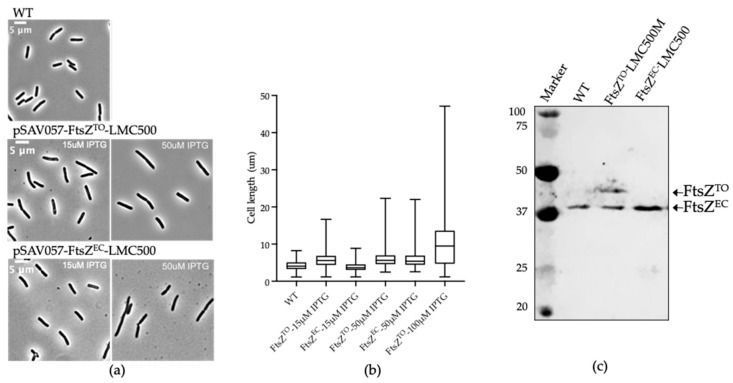
Overexpression of FtsZ^TO^ and FtsZ^EC^. (**a**) Phase contrast images of *E. coli* wild-type strain LMC500 that expressed FtsZ^TO^ or FtsZ^EC^ from plasmids pJW11 or pJW12, respectively, induced with 15, 50, or 100 μM IPTG grown in TY medium at 28 °C for 4 h. (**b**) Quantification of the length (μm) of cells treated as shown in (**a**) and of cells where FtsZ^TO^ was induced by incubation in 100 µM IPTG. The numbers of cells analyzed were, from left to right, 910, 1193, 1056, 1227, 995, and 974. (**c**) Western blot of FtsZ^TO^ and FtsZ^EC^ that were expressed in LMC500 from plasmids pJW11 or pJW12 with 15 μM IPTG induction.

**Figure 2 ijms-23-03016-f002:**
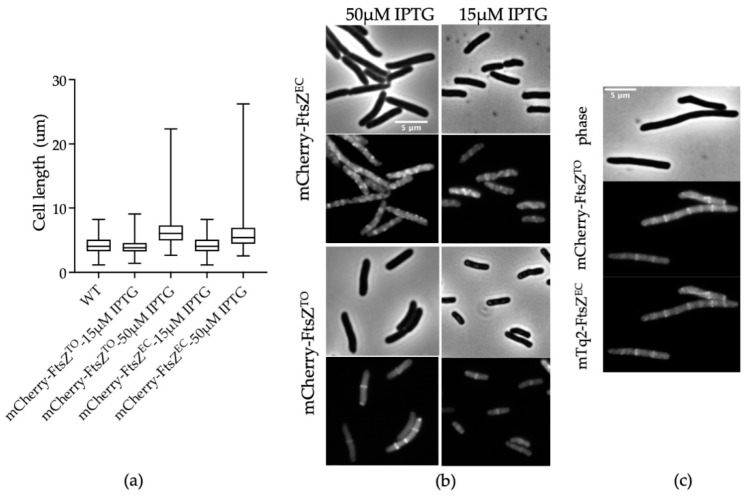
Localization of FtsZ^TO^ and FtsZ^EC^. (**a**) Cell length (μm) quantification of images shown in (**b**). The numbers of cells analyzed were, from left to right, 589, 996, 861, 901, and 995. (**b**) Phase contrast (upper) and fluorescence (lower) images of LMC500 with mCherry-FtsZ^TO^ or mCherry-FtsZ^EC^ expression from plasmid pJW13 or pJW14, respectively, induced with 15 or 50 μM IPTG for 4 h grown in TY medium at 28 °C. (**c**) Colocalization of mCherry-FtsZ^TO^ and mTq2-FtsZ^EC^. Phase contrast and fluorescence images of LMC500 with both mCherry-FtsZ^TO^ and mTq2-FtsZ^EC^ expression from plasmid pJW13 and pNM046, respectively, induced with 15 μM IPTG for 4 h grown in TY medium at 28 °C.

**Figure 3 ijms-23-03016-f003:**
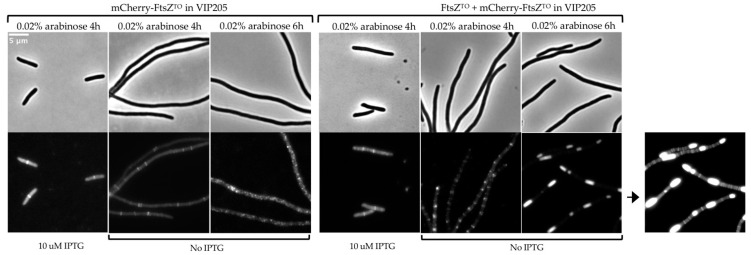
FtsZ^TO^ forms rings and barrel structures in VIP205. Phase contrast (**upper**) and fluorescence (**lower**) images of LMC500 with mCherry-FtsZ^TO^ expression (**left**) from plasmid pJW16 or with both FtsZ^TO^ and mCherry-FtsZ^TO^ expression (**right**) from plasmid pJW15, induced with 0.02% arabinose for 4 h or 6 h grown in TY medium at 28 °C.

**Figure 4 ijms-23-03016-f004:**
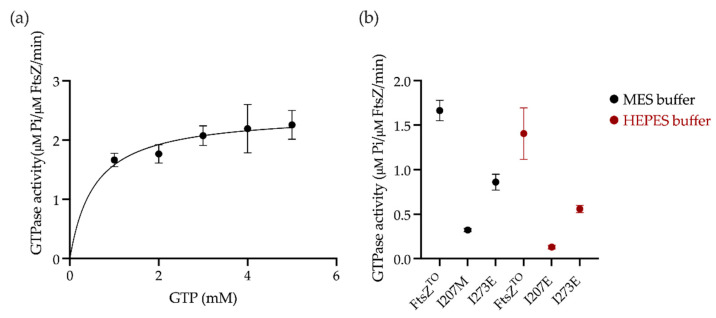
GTPase activities of FtsZ^TO^ and its variants in vitro. (**a**) Steady-state kinetic analysis of FtsZ^TO^ at 28 °C. The Michaelis–Menten equation was used to fit the data of the GTPase assay. Error bars are the standard error of the mean, n = 3 independent experiments. K_M_, k_cat_, and V_max_ values are listed in Table 1. (**b**) Comparison of GTPase activities of FtsZ^TO^ and its mutants was carried out in MES buffer pH 6.5 or HEPES buffer pH 7.5, 1 mM GTP, at 28 °C. Error bars are the standard error of the mean, n = 3 independent experiments.

**Figure 5 ijms-23-03016-f005:**
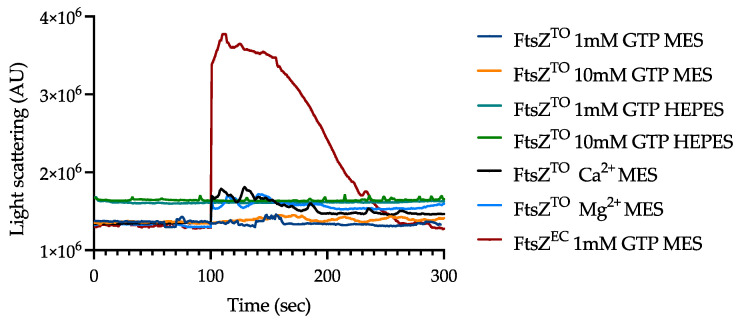
Polymerization of FtsZ^TO^ and FtsZ^EC^ assessed by 90°-angle light scattering. The background scattering was first determined for 100 s after which GTP or extra cations were added to initiate polymerization. HEPES and MES buffer had a pH of 7.5 and 6.5, respectively. The amount of Mg^2+^ was 5 mM in the buffer if no cation is mentioned. The amount Ca^2+^ and Mg^2+^ was 5 mM in FtsZ^TO^ Ca^2+^ MES and the amount of Mg^2+^ was 10 mM in FtsZ^TO^ Mg^2+^ MES. The addition of extra cations and GTP did not change the light scattering signal compared with extra cations only.

**Figure 6 ijms-23-03016-f006:**
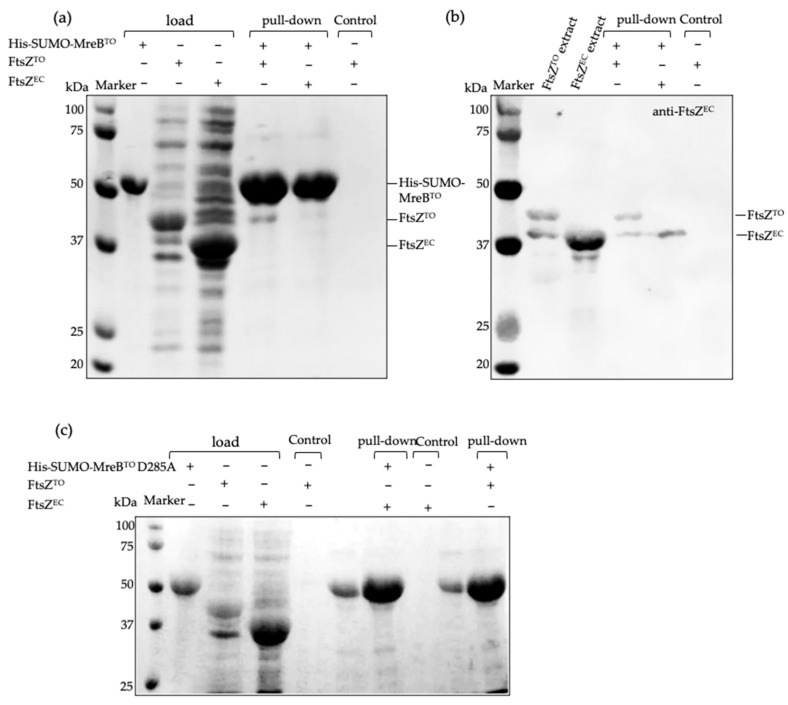
MreB^TO^ binds to FtsZ^TO^ in vitro. (**a**) Pull-down assay in vitro using His-tagged SUMO-MreB^TO^ bound to Ni-NTA agaroses and FtsZ^TO^ in solution, detected by Coomassie Blue staining. (**b**) Western blot of the proteins in pull-down assay shown in (**a**) using anti-FtsZ^EC^. (**c**) Pull-down assay in vitro using D285A variant of His-tagged SUMO-MreB^TO^ bound to Ni-NTA agaroses and FtsZ^TO^ in solution, detected by Coomassie Blue staining.

**Figure 7 ijms-23-03016-f007:**
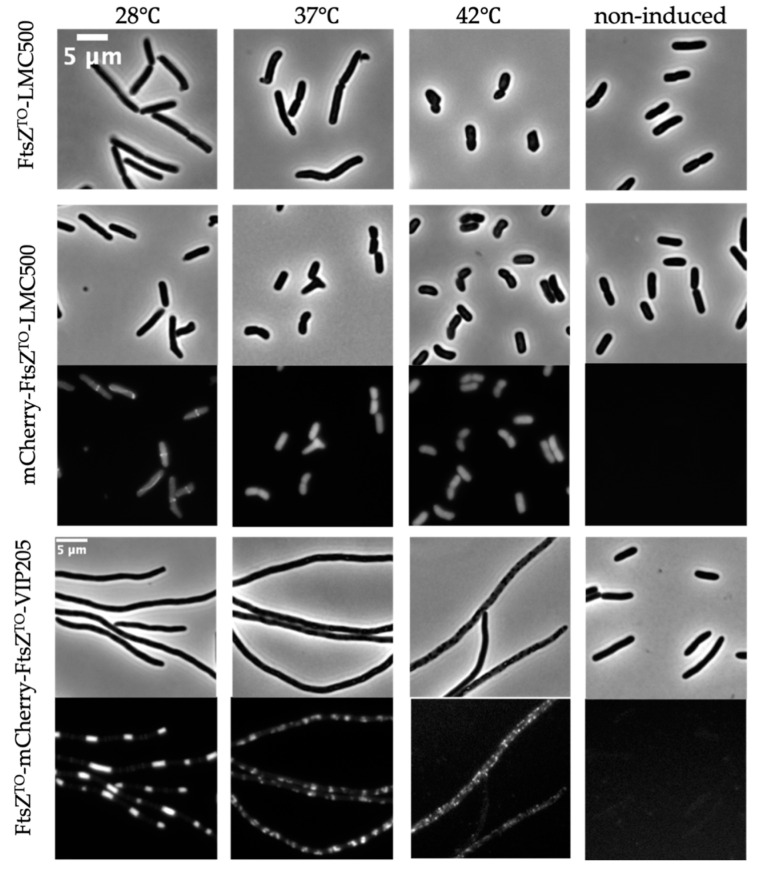
FtsZ^TO^ shows temperature-sensitive characteristics in *E. coli*. Phase contrast images of LMC500 with FtsZ^TO^ expression from plasmid pJW11 induced with 50 μM IPTG for 4 h grown in TY medium at 28 °C, 37 °C, or 42 °C (**upper panel**). Phase contrast and fluorescence images of LMC500 with mCherry-FtsZ^TO^ expression from plasmid pJW13 induced with 50 μM IPTG for 4 h grown in TY medium at 28 °C, 37 °C, or 42 °C (**middle panel**). Phase contrast and fluorescence images of VIP205 with both FtsZ^TO^ and mCherry-FtsZ^TO^ expression from plasmid pJW15 induced with 0.02% arabinose for 6 h grown in TY medium at 28 °C, 37 °C, or 42 °C (**bottom panel**).

**Table 1 ijms-23-03016-t001:** GTPase activity of FtsZ^TO^.

K_M_ (mM)	V_max_ (μM GTP/μM FtsZ/min)	k_cat_ (min^−1^)
0.55 ± 0.11	2.45 ± 0.24	0.49 ± 0.05

**Table 2 ijms-23-03016-t002:** *E. coli* strains and plasmids used in this study.

*E. coli* Strain	Relevant Genotype	Reference or Source
LMC500	MC4100 *lysA*	[54]
VIP205	F-, *araD139*, *Δ(ara-leu) 7697*, *Δ(lac)X74*, *galE15*, *galK16*, *rpsL150*, *ftsA*::*kan*-*Tu*-*lac9*-*ptac*-*ftsZ20*	[32]
BL21DE3 plysS	F-, *omp*T, *hsdS_B_* (rB^−^, mB^−^), *gal dcm* (DE3)	Invitrogen
plasmids	property	Reference or source
pJW11	*pTrc99A* down, expressing FtsZ^TO^, p15 origin, catR	This work
pJW12	*pTrc99A* down, expressing FtsZ^EC^, p15 origin, catR	This work
pJW13	*pTrc99A* down, expressing mCherry-FtsZ^TO^ fusion, pBR322 origin, ampR	This work
pJW14	*pTrc99A* down, expressing mCherry-FtsZ^EC^ fusion, pBR322 origin, ampR	This work
pJW15	pBAD vector, expressing FtsZ^TO^ and mCherry-FtsZ^TO^ fusion in one operon, ampR	This work
pJW16	pBAD vector, expressing mCherry-FtsZ^TO^ fusion, ampR	This work
pJW17	pET11b vector (Novagen), expressing FtsZ^TO^	This work
pJW18	T7 promoter, expressing His tagged SUMO-MreB^TO^ fusion, pBR322 origin, ampR	This work
pJW19	T7 promoter, expressing His tagged-SUMO-MreB^TO^ D285A fusion, pBR322 origin, ampR	This work
pJW20	pET11b vector (Novagen), expressing FtsZ^TO^ I207M	This work
pJW21	pET11b vector (Novagen), expressing FtsZ^TO^ I273E	This work
pNM046	*pTrc99Adown* down, expressing mTq2-FtsZ^EC^ fusion, pBR322 origin, ampR	[55]

## Data Availability

The data that support the findings of this study are available from the corresponding author upon reasonable request.

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
