# Peer review of "The Longitudinal Dividing Bacterium Candidatus Thiosymbion Oneisti Has a Natural Temperature-Sensitive FtsZ Protein with Low GTPase Activity"

_ijms, 2022, doi:10.3390/ijms23063016_

Round 1

Reviewer 1 Report

The authors bring interesting results about FtsZ from the longitudinally dividing Ca. T. oneisti. The main question this manuscript has ambition to answer is whether longitudinal division is caused by some unique characteristics of the FtsZTO protein. Overall quality of the manuscript is high and authors employed wide range of different methods. Unfortunately, because it is not possible to cultivate Ca. T. oneisti cells in the laboratory, the authors must rely on in vivo experiments with heterologous FtsZTO protein expression in E. coli and on in vitro experiments.  I have several questions/suggestions:

  1. Second chapter and figure 2 are bit confusing.
  • Are in Fig 2B first and second panels from top expressing mCherry-FtsZEC ? In case yes, the label is missing.
  • Line 137 (Figure 2-c) should it not be (Figure 2-b)?
  • Lines 135-138 “Since the mCherry- FtsZEC signal showed a higher fluorescent background and formed more multiple rings along cells compared to mCherry-FtsZTO (Figure 2-c), we hypothesise that mCherry FtsZTO interacts with native FtsZEC in coli to form the ring but is not under E. coli Z-ring’s dynamic spatiotemporal regulation.”- Do I understand correctly, that the upper two panels (bright field and fluorescent image) in Fig 2b show cells expressing mCherry-FtsZEC on otherwise wt background and lower two panels show cells expressing mCherry FtsZTO? If so, how can mCherry FtsZTO be responsible for multiple rings in cells that express wt FtsZEC mCherry- FtsZEC? I’m sorry, if I misunderstood the author’s intention but it is because of the confusing image labelling. Moreover, from figure legend one can think that in upper two panels we see mCherry FtsZTO and in lower two mCherry- FtsZEC, as it is stated in this order.
  • I suggest to rewrite this chapter as well the figure legend so it is more clear to the reader.
  • The hypothesis of FtsZTO and FtsZEC interaction can be tested easily.
  1. Lines 171-172 „Knowing that mCherry FtsZ can only polymerize if untagged FtsZ is present, only mCherry-FtsZTO was induced with 0.02% arabinose in VIP205 and no FtsZTO rings or barrels were observed any more.” I suggest simplifying this sentence to „When only mCherry-FtsZTO was induced with 0.02% arabinose in VIP205, no FtsZTO rings or barrels were observed.”
  2. 3 - I‘m missing image of VIP205 carrying pJW15 in the presence of 0.02% arabinose and 10 μM IPTG, which is mentioned in the text. There are only images of cells with pJW16 in the presence of arabinose and with or without IPTG but cells with JW15 are only without IPTG. Please add also this image into figure 3 to support the claim in the main text (line 160).
  3. Please, can authors be more specific about the mCherry-FtsZTO barrel like structures observed after 6 h in VIP205 with pJW15. Do authors think they are inclusion bodies?
  4. Figure 4-I suggest to indicate also pH of the buffers.
  5. In chapter 4 the authors state that the GTPase activity of FtsZTO is several times lower than activity of FtsZEC. Did authors test at least two different FtsZTO purification batches and/or different purification approaches? From my experience it is common that FtsZ GTPase activity differs from batch to batch. As the FtsZTO protein also did not sediment very well, it might not be an active batch.
  6. Is there a significant difference between the background (GTP alone) and protein in Pi measured (Fig 4a)? I suggest to add this control into the graph in the future. Or was background deducted from the values measured for samples with proteins?
  7. Besides testing different purification batches, I also suggest to add GTPase activity measurements at different time points, e.g. 10 min, 30 min, 60 min. After longer time the differences between FtsZTO and FtsZEC might decrease if are authors convinced that the protein batch is OK and the GTPase activity is still low.
  8. The chapters 1-3 indicate that FtsZTO is capable of polymerization. Moreover, the similarity between FtsZTO and FtsZEC proteins, especially within the GTPase binding residues suggest these two proteins should have similar GTPase activities and polymerization properties. However, in the Chapter 5 the authors bring evidences that FtsZTO is not polymerizing. This also can suggest that authors might be working with protein batch that is not active enough. As a consequence of the lack of FtsZTO protein polymerization the lower GTPase activity can be observed. This can be excluded by testing different purification batches and optimizing the purification process. Did authors try that with similar results?
  9. Fig S2 – Please, can authors add in the panel with Coomasie also lane with FtsZEC which is visible in WB and Ponceau panels?
  10. In legend of Fig S4 there are typos “HEPSE buffer”.

Author Response

We have added a word file in which each point raise by reviewer 1 is commented on and as far as we are aware all issues were solved.

Reviewer 2 Report

 The manuscript entitled “The longitudinal dividing bacterium Candidatus Thiosymbion  oneisti has a natural temperature sensitive FtsZ protein with  low GTPase activity “ presents a thorough characterization of the FtsZ protein from the rod-shaped 12 gamma-proteobacterium Candidatus Thiosymbion oneisti when  expressed in E. coli and when studied in vitro. The experiments are carefully done, the article is clearly written and the conclusion drawn, are mostly, well founded on the experimental results.  Temperature sensitivity, the   low GTPase activity, its interaction with MreBTO and its interaction with FtsZEC are well documented, supporting the final interpretation that the behavior of FtsZTO in E. coli cells and the similarity between the two proteins points in the direction that the difference in shape and size of the FtsZTO ring is likely the result of its interaction with Z-ring organizing proteins, which is a relevant and interesting conclusion for understanding cell division mechanisms in different organisms.

I only have one small question regarding the conclusion that FtsZTO does not polymerize in vitro. Light scattering assays were performed at a fixed concentration 25 µM of FtsZTO  and varying concentration of GTP. Given that the cooperative assembly of FtsZ is well documented, can it be ruled out that the assembly could happen at different protein or KCl concentrations?  

I consider that after this minor issue is addressed, mentioned or discussed, the paper can be published as it is.

Author Response

We have added a word file in which each question raised by the reviewer was commented and as far as we are aware alls issues were solved.

Round 2

Reviewer 1 Report

I’d like to thank the authors for the answers to my questions and implementations of my suggestions. My questions have been answered satisfactory and I find the results very interesting. However, I suggest that some of the answers should be adapted and involved in the paper prior publishing.

1. I suggest to adapt and add details about GTPase activity from the answer below into the corresponding chapter in Results (or Materials and methods):

“For each GTPase measurement, a new FtsZ batch was isolated. The GTPase of isolated FtsZ does not decrease for short time period like 1 h. The free phosphate release from GTP was measured at seven different reaction time points (from 4 min up to 30 min) to calculate the GTPase activity of FtsZTO and its variants.”

2. Sharing at least some data from testing polymerization activity under different polymerization conditions with several batches of FtsZ protein, concentration of FtsZ, buffer types, different temperatures, etc., showing no improvement of its sedimentation in the corresponding part in the Supplementary material (Fig. S5) would be of interest for the readers.

3. In legend of Fig S4 there is one more “HEPSE buffer” typo (line 110).

Author Response

A file is uploaded that covers all the raised issues by the reviewer in this second round.
